# Licochalcone A Exerts Anti-Cancer Activity by Inhibiting STAT3 in SKOV3 Human Ovarian Cancer Cells

**DOI:** 10.3390/biomedicines11051264

**Published:** 2023-04-24

**Authors:** Jeonghyeon Seo, Da Eun Lee, Seong Mi Kim, Eunjung Kim, Jin-Kyung Kim

**Affiliations:** 1Department of Biomedical Science, Daegu Catholic University, Gyeongsan-si 38430, Republic of Korea; 2Department of Food Science and Nutrition, Daegu Catholic University, Gyeongsan-si 38430, Republic of Korea

**Keywords:** Licochalcone A, ovarian cancer cells, apoptosis, cell cycle arrest, STAT3

## Abstract

Licochalcone A (LicA), a major active component of licorice, has been reported to exhibit various pharmacological actions. The purpose of this study was to investigate the anticancer activity of LicA and detail its molecular mechanisms against ovarian cancer. SKOV3 human ovarian cancer cells were used in this study. Cell viability was measured using a cell counting kit-8 assay. The percentages of apoptotic cells and cell cycle arrest were determined by flow cytometry and Muse flow cytometry. The expression levels of proteins regulating cell apoptosis, cell cycle, and the signal transducer and activator of transcription 3 (STAT3) signaling pathways were examined using Western blotting analysis. The results indicated that LicA treatment inhibited the cell viability of SKOV3 cells and induced G2/M phase arrest. Furthermore, LicA induced an increase in ROS levels, a reduction in mitochondrial membrane potential, and apoptosis accompanied by an increase in cleaved caspases and cytoplasmic cytochrome c. Additionally, LicA caused a dramatic decrease in STAT3 protein levels, but not mRNA levels, in SKOV3 cells. Treatment with LicA also reduced phosphorylation of the mammalian target of rapamycin and eukaryotic translation initiation factor 4E-binding protein in SKOV3 cells. The anti-cancer effects of LicA on SKOV3 cells might be mediated by reduced STAT3 translation and activation.

## 1. Introduction

Ovarian cancer is a cancerous tumor of the ovary that can develop in the ovary itself or communicate with surrounding structures, such as fallopian tubes and the lining of the abdomen, with the latter being more common. Ovarian cancer involving the reproductive system can have a significant impact on a woman’s quality of life. Based upon GLOBOCAN estimates, female ovarian cancer was the eighth most diagnosed malignancy globally in 2020, accounting for 3.4% of all cancer diagnoses. In addition, ovarian cancer was the eighth greatest cause of global cancer deaths in women, accounting for 4.7% of deaths [1]. In Korea, the number of ovarian cancer patients in 2019 was 24,134, an increase of 33.2% compared to 18,115 in 2016. According to the ‘2019 National Cancer Registry Annual Report’, the mortality rate of ovarian cancer in 2019 was 42.7%, which was much higher than those of other female cancers, such as breast cancer (10.6%), cervical cancer (27.4%), and endometrial cancer (10.9%), in Korea [2]. One of the reasons for the high mortality rate of ovarian cancer might be because ovaries are located in the abdominal cavity, meaning that most early ovarian cancers progress without producing symptoms. About 70% of ovarian cancers are not discovered until they have progressed beyond stage 3 [3,4]. The treatment for ovarian cancer is, similarly to other cancers, usually surgery or chemotherapy. However, after these treatments, patients often succumb to recurrence with chemotherapeutic resistance within several years after the initial treatment; therefore, new treatments and targeted therapy are essential for the effective treatment of ovarian cancer [3,4].

Signal transducer and activator of transcription 3 (STAT3) is a key transcription factor that regulates gene expression related to inflammation, cell transformation, survival, proliferation, invasion, angiogenesis, and metastasis in cancer [5,6,7]. Significant evidence has highlighted the importance of aberrantly activated STAT3 signaling in a variety of cancers, including ovarian cancer cell lines and tissue samples, as detected by microarray analysis, real-time reverse transcription-PCR, Western blot, and the luciferase reporter assay [8,9,10]. Taking advantage of the fact that STAT3 is overexpressed or hyperactivated in ovarian cancer cells, inhibiting STAT3 activation can dramatically suppress tumor growth, suggesting that STAT3 signaling is a promising target for ovarian cancer therapy [11,12,13]. Additionally, targeting the STAT3 signaling pathway has emerged as a promising therapeutic strategy for numerous cancers. Thus, it is crucial to search for STAT3 inhibitors in order to develop effective therapeutic interventions for ovarian cancer.

Licochalcone A (LicA), a major phenolic constituent of the licorice species *Glycyrrhiza inflata*, with a molecular formula of C_21_H_22_O_4_, has shown broad anti-inflammatory, antibacterial, anti-cancer, and antioxidative bioactivities [14]. Our team has been conducting research on LicA for many years, paying attention to its biological and physiological activities. Indeed, our previous study has shown that LicA treatment can block tumor formation in an azoxymethane/dextran sodium sulfate-induced colon cancer mouse model [15]. LicA can also reduce hepatic metastases in murine models of hepatic colon metastases [15]. In addition, we have found that LicA can block angiogenesis, which is crucial for cancer development and progression, both in vitro and in vivo [16]. So far, there have been two reports verifying the effect of LicA on human ovarian cancer cell lines. One is that LicA promotes apoptosis by stimulating the production of reactive oxygen species [17], and the other is that LicA exerts its anti-cancer activity by promoting death receptor expression and mitochondria-mediated apoptosis [18]. Although it has been found that the anti-cancer effects of LicA against various human cancer cells are deeply associated with induction of apoptosis, cell cycle arrest, and autophagy [14,19,20,21], the anti-cancer activity of LicA on human ovarian cancer cells and related mechanisms has not been clearly identified. In this study, the effects of LicA on proliferation, apoptosis, and cell cycle arrest of SKOV3 human epithelial ovarian cancer cells were determined and a possible mechanism involved in anti-cancer activity of LicA against ovarian cancer through STAT3-mediation was identified.

## 2. Materials and Methods

### 2.1. Materials

All reagents, including LicA, were purchased from Sigma-Aldrich (St. Louis, MO, USA) unless otherwise noted. Cycloheximide and MG262 were obtained from Thermo Fisher Scientific (Waltham, MA, USA) and Enzo Life Sciences (Farmingdale, NY, USA), respectively. Antibodies (Abs) used in this study included Cyclin A (sc-751), Cyclin B1 (sc-752), cell division cycle 2 (CDC2, sc-54), CDC25C (sc-327), proliferating cell nuclear antigen (PCNA, sc-54), Cytochrome C (sc-7159), and B-cell lymphoma-extra-large (Bcl-xL, sc-7195) antibodies obtained from Santa Cruz Biotechnology (Santa Cruz, CA, USA); cleaved caspase-3 (#9664), cleaved caspase-7 (#9491), cleaved caspase-8 (#9496), cleaved caspase-9 (#9501), cleaved Poly (ADP-ribose) polymerase (PARP, #9541), BAX (#2772), Bcl-2 (#2856), phosphor-STAT3 (#9145), and STAT3 (#9139) were obtained from Cell Signaling Technology (Danvers, MA, USA); and GAPDH (MA5-15738) was obtained from Invitrogen (Waltham, MA, USA). PE mouse anti-human DR4 (#12-644-42) and PE mouse anti-human Fas ligand (FasL, # 564261) were obtained from Invitrogen (Carlsbad, CA, USA) and BD Biosciences (Bedford, MA, USA), respectively.

### 2.2. Cells and Cell Viability Assay

SKOV3 human epithelial ovarian cancer cells were purchased from the Korea Cell Bank (Seoul, Korea). They were cultured in RPMI-1640 (Hyclone, Logan, UT, USA) supplemented with heat-inactivated fetal bovine serum (FBS, Hyclone) and a penicillin–streptomycin mixed solution (Hyclone) at 37 °C with 5% CO_2_. LicA was dissolved in dimethyl sulfoxide (DMSO, 40 mM) and filtered with a 0.22 μm microfilter. Then, the working solutions of LicA were diluted with RPMI-1640 to final concentrations. The final concentration of DMSO in solution was less than 0.1%. Cells were seeded into 96-well plates at a density of 5 × 10^3^ cells/well and incubated at 37 °C with 5% CO_2_ for 24 h. Cells were treated with LicA for 24 h. Thereafter, 10 μL CCK-8 (Dojindo, Tokyo, Japan) was added to each well followed by incubation at 37 °C with 5% CO_2_ for another 2 h. Absorbance at 450 nm was then measured with a microplate reader (Tecan, San Jose, CA, USA).

### 2.3. Measurement of Apoptotic Cells

Apoptosis was assayed using a Muse™ Annexin V and Dead Cell Kit (Luminex, Austin, TX, USA) according to the manufacturer’s user guide. SKOV3 cells grown in 24-well plates (5 × 10^4^ cells/well) were treated with LicA at different concentrations for 24 h. Adherent cells were washed, collected, and incubated with Annexin V and 7-AAD (7-Aminoactinomycin D-dead cell marker) for 20 min at room temperature in the dark. Percentages of live, dead, early, and late apoptotic cells were measured with a Muse^®^ Cell Analyzer (Luminex).

### 2.4. Determination of Mitochondrial Membrane Potential (MMP)

SKOV3 cells (1 × 10^5^ cells) were seeded into 12-well cell culture plates and treated with LicA at different concentrations for 6 h. Following harvest, cells were centrifuged at 4000 rpm for 5 min at 4 °C and washed with 1× assay buffer. In a dark condition, cells were added with Mitopotential working solution (Luminex) and incubated at 37 °C for 20 min. After adding 7-AAD, cells were incubated at room temperature for 10 min. MMP was then assayed using a Muse™ cell analyzer (Luminex).

### 2.5. Measurement of Reactive Oxygen Species (ROS) Production

ROS levels were quantified in accordance with the protocol of a Muse™ Oxidative Stress Kit (Luminex). The Muse^®^ Oxidative Stress Kit allows for quantitative measurement of ROS, namely superoxide radicals in cells undergoing oxidative stress. Cells (1 × 10^5^ cells per well) in 12-well plates were treated with LicA at indicated concentrations for 6 h and stained with Muse™ Oxidative Stress reagent at 37 °C for 30 min. Cells were then collected for ROS analysis using a Muse^®^ Cell Analyzer (Luminex).

### 2.6. Western Blot Analysis

SKOV3 cells (2 × 10^6^ cells) grown in culture medium on 100 mm plates were treated with LicA at indicated concentrations for 24 h. Cells were rinsed with ice-cold PBS, followed by addition of 100 μL of PRO-PREP™ protein extraction solution (Seongnam, Korea) containing a fresh mixture of complete protease and phosphatase inhibitors (Roche, Switzerland) to each sample. After proteins (30 μg) were separated by 12% sodium dodecyl sulfate-polyacrylamide gel electrophoresis, they were transferred to a PVDF membrane (ATTO, Tokyo, Japan). Target proteins were immunoblotted with indicated primary antibodies (1:1000 dilution) at 4 °C overnight and then incubated with an HRP-conjugated secondary antibody (1:5000 dilution) at 4 °C for 3 h. Protein bands were then visualized using an enhanced chemiluminescent reagent (Thermo Fisher Scientific) and DAVINCH Chemi CAS-400SM (Davinch-k, Seoul, Korea). Protein levels were analyzed using a TotalLab analysis software (Nonlinear Dynamics, Durham, NC, USA).

### 2.7. Real-Time Reverse Transcription Polymerase Chain Reaction (RT-qPCR)

Total RNAs were isolated from colon homogenates with an RNAeasy mini kit (Qiagen, Hilden, Germany) according to the manufacturer’s instructions. After RNA preparation, cDNAs were synthesized with a PrimeScript™ RT Reagent Kit and gDNA Eraser (TakaRa, Shiga, Japan). Subsequently, mRNA expression levels were quantified using cDNA and a LightCycler^®^ 96 Real-Time PCR System (Roche Diagnostics AG, Rotkreuz, Switzerland) with a Lightcycler 480 SYBR Green I Master (Roche Diagnostics AG). mRNA expression levels were normalized to GAPDH and relative mRNA expression was calculated as fold change. Primers used in qPCR were STAT3, forward 5′-AGA AGG ACA TCA GCG GTA AG-3′ and reverse 5′-CCT TGG GAA TGT CAG GAT AGA G-3′; GAPDH, forward 5′-TGC ACC ACC AAC TGC TTA GC-3′ and reverse 5′-GGC ATG GAC TGT GGT CAT GAG-3′.

### 2.8. Statistical Analysis

Values are expressed as mean ± SEM of results from at least three experiments. Comparisons were performed by one-way analysis of variance (ANOVA). Bonferroni’s correction for multiple comparisons was used to determine the level of significance. Statistical significance was considered at *p* < 0.05.

## 3. Results

### 3.1. LicA Inhibits Proliferation of SKOV3 Cells by Inducing Apoptosis

To investigate whether LicA treatment could inhibit the proliferation of ovarian cancer cells, SKOV3 cells were treated with LicA at different concentrations for 24 h. LicA at concentrations ≥10 μM significantly blocked the proliferation of SKOV3 cells in a dose-dependent manner, as shown in Figure 1b. The half maximal inhibitory concentration value (IC50) calculated from dose–response curves for LicA was 19.22 μM at 24 h (Appendix A).

Next, we determined the contribution of programmed cell death to loss of viability of SKOV3 cells by measuring apoptosis after staining cells with Annexin V and 7-AAD. Apoptosis was not induced by LicA at a concentration of 5 μM, which did not affect cell proliferation. However, apoptosis, especially late apoptosis, was significantly increased in SKOV3 cells treated by LicA at 10 or 25 μM (Figure 2a). In subsequent experiments, SKOV3 cells were treated with LicA at concentrations higher than 10 μM as such concentrations had an antiproliferative effect on SKOV3 cells.

To investigate effects of LicA on caspase cascade in the apoptotic signaling pathway, cleaved forms of effector caspases and its upstream initiators caspases were measured with Western blotting. After SKOV3 cells were treated with 10 or 25 μM of LicA for 24 h, expression levels of cleaved forms of caspase 3, 7, 8, and 9 were increased by LicA in a concentration-dependent manner (Figure 2b). In addition, the level of cleaved PARP was increased by LicA treatment for 24 h (Figure 2b). It is well known that PARP, one of DNA repair enzymes, will be cleaved by caspase-3 to inactivate it in cells undergoing apoptosis [22]. Expression levels of DR4 and FasL on the cell surface of SKOV3 cells were also reduced by LicA treatment, as determined by flow cytometry (Figure 2c). These findings suggest that LicA can induce SKOV3 cell apoptosis by activating both extrinsic caspase 8- and intrinsic caspase 9-mediated pathways and their downstream effectors caspases 3 and 7.

### 3.2. LicA Reduces Mitochondrial Membrane Potential in SKOV3 Cells

To further investigate whether dysfunction of mitochondria occurred in LicA-induced apoptosis, mitochondrial membrane potential (MMP) was measured using a Muse MitoPotential Kit. As shown in Figure 3a, the percentage of depolarized cells was 46.7% in the group treated with 10 μM of LicA in comparison with the control (7.5%). This result indicates that LicA treatment can enhance permeability of the mitochondria membrane and cause dissipation of MMP in SKOV3 cells.

Since the permeability of mitochondrial membrane was enhanced by LicA, expression levels of Bcl-2 family proteins were determined by Western blot analysis. As shown in Figure 3b, anti-apoptotic proteins, Bcl-2 and Bcl-xL, in SKOV3 cells were downregulated by treatment with LicA. In contrast, the expression level of a pro-apoptotic protein BAX was increased by treatment with LicA (Figure 3b). Consistent with these results, the levels of cytochrome c were enhanced by LicA in a concentration-dependent manner (Figure 3b). These results suggest that one of the mechanisms of action involved in the apoptosis induced by LicA is mitochondrial-mediated apoptosis.

### 3.3. LicA Suppresses Intracellular ROS Generation

Intracellular ROS play a vital role in different types of cell survival. ROS accumulation can induce cell death in many types of cancer cells after treatment with anti-cancer agents [23]. Therefore, we treated cells with different concentrations (0, 10, and 25 μM) of LicA for 24 h and analyzed the population of ROS positive cell using a Muse cell analyzer to measure intracellular ROS induced by LicA in SKOV3 cells. As shown in Figure 4, the amount of ROS was increased by LicA in a concentration-dependent manner. LicA treatment increased ROS positive cells from 17.1 ± 1.9% in the control to 43.1 ± 4.8% in 25 μM of LicA-treated SKOV3 cells (Figure 4).

### 3.4. LicA Induces G2/M Phase Cell Cycle Arrest in SKOV3 Cells

To determine the effects of LicA on cell cycle in SKOV3 cells, DNA content was measured after treatment with LicA for 24 h. The obtained results suggested that LicA could arrest the cell cycle at G2/M phase, as shown in Figure 5a,b. Briefly, 25.2 ± 1.3% of control SKOV3 cells were gated into G2/M phase. After treatment with 25 μM LicA, percentage of cells in the G2/M phase was increased to 34.4 ± 1.1%. LicA also increased the percentage of cells at sub-G1 phase (2.8 ± 0.2% in control cells to 3.3 ± 0.3% in 25 μM LicA-treated cells, Figure 5a), which indicated apoptotic cells. This indicates that LicA can cause apoptosis and arrest the cell cycle at the G2/M phase in SKOV3 cells.

To determine whether expression levels of cyclin A, cyclin B, CDC2, and CDC25C responsible for G2/M regulation were affected by LicA treatment, expression levels of these proteins were measured by Western blotting. Expression levels of cyclin A and cyclin B were increased in LicA-treated cells compared to those in control cells (Figure 5b). However, expression levels of CDC2 and CDC25C were significantly decreased in 25 μM LicA-treated SKOV3 cells compared to those in control cells (Figure 5b). These results suggest that CDC2 and CDC25C might play key roles in LicA-caused G2/M phase arrest.

### 3.5. LicA Blocks STAT3 Phosphorylation and Expression

STAT3 is broadly hyperactivated in various cancer cells. It can promote oncogenesis by up-regulating the expression of target proteins associated with cell cycle progression and apoptosis [5,6]. Thus, we investigated the effects of LicA on phosphorylation and expression of STAT3 protein in SKOV3 cells. Results showed that LicA inhibited STAT3 phosphorylation and protein expression in SKOV3 cells in a concentration-dependent manner (Figure 6a). In particular, phosphorylated STAT3 was hardly detected in SKOV3 cells treated with 10 or 25 μM LicA. Simultaneously, the amounts of STAT3 protein in the nucleus and cytoplasm were significantly decreased in LicA-treated group compared to those in the control group (Figure 6b).

Since LicA treatment significantly reduced protein levels of STAT3, we next examined mRNA levels of STAT3 in SKOV3 cells. Interestingly, mRNA levels of STAT3 were not decreased by LicA treatment in SKOV3 cells, but rather showed a tendency to increase, although the increase was not statistically significant (Figure 6c). These results led us to investigate how LicA reduced STAT3 protein independent of mRNA expression. To first determine whether LicA affected proteasome-mediated degradation of STAT3 protein, proteasome inhibitor MG262 was added and protein levels of STAT3 were checked by Western blot analysis. MG262 combined with LicA failed to restore STAT3 protein levels reduced by LicA treatment, as shown in Figure 6d. Cells treated with LicA and MG262 or CHX were compared to LicA-treated cells, but no statistically significant differences were observed. These data indicate that the proteasome might not be involved in the reduction in STAT3 levels by LicA treatment.

To further determine whether LicA-induced changes of STAT3 protein levels were associated with translational regulation, cycloheximide (CHX), a protein translation inhibitor, was utilized to prevent de novo STAT3 protein synthesis in SKOV3 cells treated with or without LicA. As shown in Figure 6d, it was confirmed that the amount of STAT3 protein was significantly reduced in the CHX-alone treatment group compared to that in the control group. However, the group treated with CHX and LicA showed no significant difference in STAT3 protein expression from the group treated with LicA alone (Figure 6d). This suggests that LicA does not inhibit STAT3 translation by inhibiting translation elongation.

Protein synthesis initializes with binding of eukaryotic initiation factor 4E (eIF4E) to the m7G cap at the 5’ end of eukaryotic mRNAs [24]. The mammalian target of rapamycin (mTOR) pathway could encourage phosphorylation of eIF4E-binding protein 1 (4EBP1) and prompt the dissociation of 4EBP1 from eIF4E, enabling the association of eIF4E and eIF4G [24]. To determine whether LicA could block protein synthesis at the level of initiation, we investigated the regulation of 4EBP1 and eIF4E. Western blot analysis showed that LicA treatment inhibited the activation of mTOR, as well as phosphorylation of 4EBP1, in a concentration-dependent manner without reducing eIF4E protein levels (Figure 6e). These results indicate that LicA can repress STAT3 expression at the translational level, especially at translation initiation.

## 4. Discussion

Ovarian cancer is the leading cause of death among all genital cancers in women. One of the reasons for the high mortality rate of ovarian cancer is that it is usually not diagnosed until an advanced stage due to its minor, vague symptoms that make appropriate treatment difficult [3,4]. Despite recent widespread awareness of ovarian cancer, survival rates have not changed significantly due to difficulties in early diagnosis. To cope with these difficulties, research studies on the appropriate treatment methods for ovarian cancer are ongoing. One of these studies is the use of phytochemicals for treatment. In fact, extensive laboratory studies have shown that phytochemicals, such as polyphenols, flavones, and flavonoids, can exert potential anti-cancer properties against various types of cancer [25,26,27,28].

LicA has demonstrated potential pharmacological properties, including its anti-cancer activity [14,15,16,17,18,19,20,21,26]. The anti-cancer effects of LicA on various types of cancer cells, including colon [29], lung [30], bladder [31], and breast cancers [17,32], have been documented. LicA can promote cell cycle arrest, induce apoptosis, and suppress angiogenesis and metastasis [14,17,19,29,30,31,32]. However, the effects of LicA on ovarian cancer cells are very limited. They have not been fully elucidated yet. For example, two studies have examined the synergistic effect of LicA in combination with other chemotherapeutic agents against ovarian cancer. Lee et al. [18] have investigated the combined effect of LicA and 3-(5′-Hydroxymethyl-2′-furyl)-1-benzyl indazole (YC-1) on cell death in ovarian cancer cells. Results showed that YC-1 could enhance LicA-induced apoptosis-related protein activation, nuclear damage, and cell death. Another study examining the effects of LicA on ovarian cancer showed that LicA could promote cell death induced by geldanamycin [17]. Based on these facts, we conducted a study to elucidate the anti-cancer activity of LicA at the molecular level in ovarian cancer cells.

Our study showed that LicA inhibited the proliferation of SKOV3 cells (Figure 1b). This result was consistent with previous findings using various human cancer cells [14,19,21,29,30,31,32]. We first clarified that growth inhibition induced by LicA was mediated by apoptosis (Figure 2 and Figure 3) and cell cycle arrest (Figure 5). Although the effect of LicA on cytotoxicity in normal ovarian epithelial cells, not cancer cells, has not been fully elucidated in the present study, previous studies using human umbilical vein endothelial cells showed that LicA did not affect cell proliferation up to 10 μM of LicA [16]. This result suggests that cancer cells are more sensitive to LicA cytotoxicity than normal cells. Many studies have demonstrated that LicA can be used as a chemopreventive agent for cancer due to its ability to induce growth inhibition, cell cycle arrest, and apoptosis in several human cancer cell lines [14,19,21,29,30,31,32]. Although the results of this study are limited by using only one type of ovarian cancer cell line (SKOV3), our results provide experimental evidence for the use of LicA as a chemopreventive agent against ovarian cancer.

Like previous studies using various cancer cells [30,31,32], LicA also induced intracellular ROS production in SKOV3 cells (Figure 4). ROS are well known as important regulators that are closely related to apoptosis. The most common way for ROS to kill cancer cells is through the activation of apoptosis. Both intrinsic and extrinsic pathways of apoptosis are known to be activated by ROS [33]. The extrinsic pathway of apoptosis is mediated by the binding of death-inducing ligands to their cognate receptors, which can recruit adapter proteins and procaspase assemble death-inducing signaling complexes and activate effector caspases [34]. This interaction is completed by the cellular FLICE-inhibitory protein (c-FLIP). ROS can downregulate c-FLIP half-life by inducing its ubiquitin-proteasomal degradation, thereby enhancing this extrinsic pathway [35]. Meanwhile, for most ROS-related anti-cancer agents, apoptosis relies on the activation of an intrinsic pathway involving mitochondrial permeability, which increases with the release of pro-apoptotic factors, such as cytochrome c, within the cytoplasm [33,34]. The released cytochrome c can create a complex with the cytosolic protease activator of apoptosis 1 and procaspase 9 to form apoptosomes, which, in turn, can activate effector caspases [34]. The current study showed that levels of cleaved caspase-8 and expression levels of DR4 and FasL on the cell surface were increased in LicA-treated SKOV3 cells (Figure 2c–f). On the other hand, loss of MMPs (Figure 3a,b), upregulated BAX/Bcl-2 ratio, and elevated levels of cytosolic cytochrome C (Figure 3c,d) were also observed in SKOV3 cells exposed to LicA. Therefore, LicA can induce apoptosis through both extrinsic and intrinsic pathways, which might be related to ROS increase by LicA treatment.

LicA treatment arrested the cell cycle at G2/M phase by downregulating CDC2 and CDC25C in SKOV3 cells (Figure 5). Previous experiments with human lung and bladder cancer cells have confirmed that LicA can cause G2/M phase arrest by inhibiting CDC2, CDC25C, cyclin A, and cyclin B expression [31,36]. Contrary to these previous results, no LicA-induced reduction in cyclin A and cyclin B levels was observed in this study. In addition, several studies have shown that LicA can arrest cell cycle at G0/G1 in glioma [21] and colon cancer cells [37], but not in G2/M. It is clear that LicA can cause cell cycle arrest in G0/G1 and/or G2/M phases, although detailed patterns and mechanisms might differ from cell to cell.

Here, it was found that LicA reduced both protein and phosphorylation levels of STAT3 (Figure 6a). Interestingly, no LicA-induced decrease in STAT3 mRNA level was observed (Figure 6c), suggesting that regulation of STAT3 by LicA occurred at post-transcriptional level. Experiments with proteasome inhibitors and translational elongation inhibitors confirmed that LicA did not inhibit protein expression through either proteasome-mediated degradation or reduced translational elongation, respectively (Figure 6d). Further examination indicated that LicA prohibited protein synthesis of STAT3 possibly through the inhibition of mTOR/4EBP1 axis (Figure 6e). The suppression of mTOR activity by ouabain led to the hypo-phosphorylation of 4EBP1 which could bind and sequester eIF4E, thus preventing the formation of eIF4 complex and blocking translation initiation [38,39,40].

Previous studies have already shown that LicA significantly inhibits the phosphorylation and nuclear localization of STAT3 in leukemia cells [41]. Additionally, Shu et al. demonstrated that LicA significantly inhibited IgE-mediated STAT3 phosphorylation in bone marrow mast cells [42]. However, in these two studies using leukemia cells and bone marrow-derived mast cells, there was no change in STAT3 protein expression by LicA treatment. Here, we proposed a novel mechanism by which LicA could inhibit ovarian cancer cell malignancy by downregulating STAT3 translation initiation in SKOV3 cells and then lowering STAT3 protein and activation levels.

Taken together, the present study demonstrates that LicA can efficiently inhibit the viability of ovarian cancer cells by inducing apoptosis via the activation of caspases and the disruption of the mitochondrial membrane potential. LicA also arrested the cell cycle at the G2/M phase. To confirm the effectiveness of this compound, further investigation is needed to determine how ovarian cancer cells respond to LicA treatment.

## Figures and Tables

**Figure 1 biomedicines-11-01264-f001:**
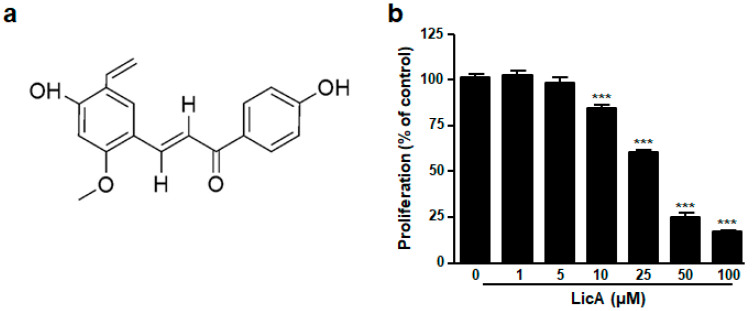
Effect of LicA on cell viability in SKOV3 cells. (**a**) Chemical structure of LicA. (**b**) Cell viability of SKOV3 cells. SKOV3 cells were treated with LicA with increasing concentrations (0, 1, 5, 10, 25, 50, and 100 μM) for 24 h. Cell viability was detected using CCK-8 assay. Data are presented as mean ± SEM. *** *p* < 0.001 compared with untreated control cells (0 μM).

**Figure 2 biomedicines-11-01264-f002:**
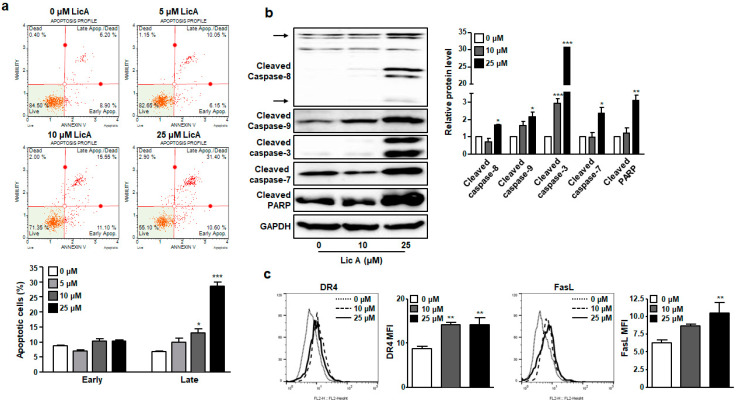
LicA induces apoptosis in SKOV3 cells. SKOV3 cells were treated with indicated concentration of LicA for 24 h. (**a**) The population of apoptotic cells was determined with a Muse™ flow cytometer. Respective results are presented. (**b**) Levels of cleaved caspase-8, cleaved caspase-9, cleaved caspase-3, and cleaved PARP in LicA-treated SKOV3 cells detected with Western blot. (**c**) Cell surface DR4 and FasL in LicA-treated SKOV3 cells were measured with flow cytometry. Data are presented as mean ± SEM. * *p* < 0.05, ** *p* < 0.01, *** *p* < 0.001, compared with untreated control cells (0 μM).

**Figure 3 biomedicines-11-01264-f003:**
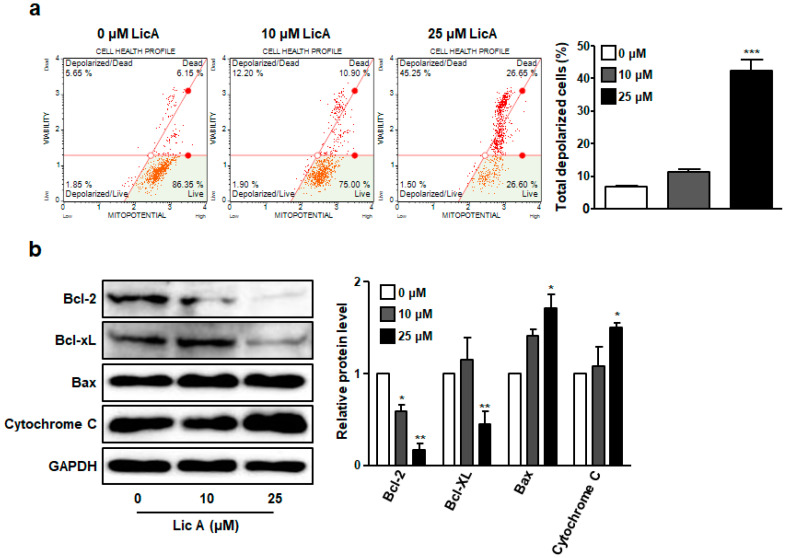
LicA increased mitochondrial depolarization in SKOV3 cells. SKOV3 cells were treated with LicA at the indicated concentration for 24 h. (**a**) Depolarization of the mitochondrial membrane was evaluated with a Muse™ flow cytometer. Graphs showing percentages of live and dead depolarized cells are indicated. (**b**) Levels of Bcl-2, Bcl-xL, BAX, and cytochrome C in LicA-treated SKOV3 cells detected with Western blot. Data are presented as mean ± SEM. * *p* < 0.05, ** *p* < 0.01, *** *p* < 0.001 compared with untreated control cells (0 μM).

**Figure 4 biomedicines-11-01264-f004:**
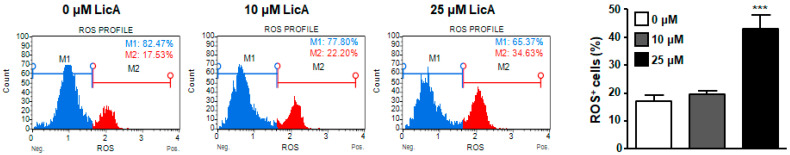
LicA increases intracellular ROS levels in SKOV3 cells. SKOV3 cells were treated with LicA at the indicated concentrations for 6 h. Percentage of ROS generation plots of Muse Cell Analyzer results. Data are presented as mean ± SEM. *** *p* < 0.001 compared with untreated control cells (0 μM).

**Figure 5 biomedicines-11-01264-f005:**
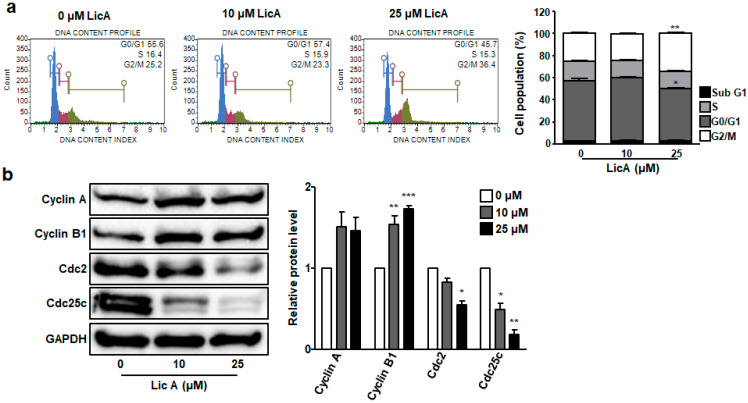
LicA induces cell cycle arrest at G2/M phase in SKOV3 cells. SKOV3 cells were treated with LicA at indicated concentrations for 24 h. (**a**) Cell cycle histogram of representative results of LicA-treated SKOV3 cells was obtained using a Muse Cell Cycle kit and a Muse Cell Analyzer. Percentages of different cell populations are indicated as a graph. (**b**) Levels of cyclin A, cyclin B1, CDC2, and CDC25C in LicA-treated SKOV3 cells were detected with Western blot. Data are presented as mean ± SEM.* *p* < 0.05, ** *p* < 0.01, *** *p* < 0.001, compared with untreated control cells (0 μM).

**Figure 6 biomedicines-11-01264-f006:**
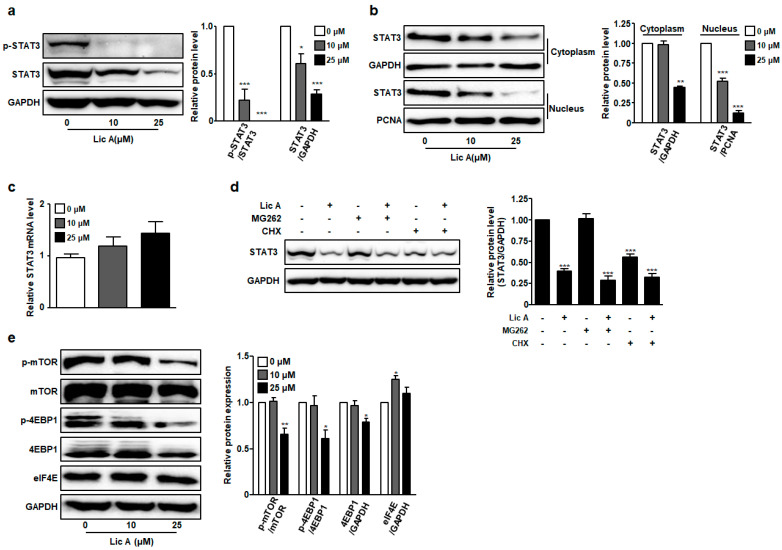
LicA inhibits STAT3 translation in SKOV3 cells. SKOV3 cells were treated with LicA at the indicated concentrations for 24 h. (**a**) Expression levels of p-STAT3 and STAT3 were detected by Western blotting. (**b**) Protein expression levels of STAT3 in the cytoplasmic and nuclear fractions were detected by Western blotting (GAPDH as a cytoplasmic marker and PCNA as a nuclear marker). (**c**) mRNA expression of STAT3 was determined by qRT-PCR. (**d**) SKOV3 cells were treated with vehicle or LicA (25 μM) in the presence or absence of cycloheximide (CHX) or MG262. Expression levels of STAT3 were then analyzed by Western blotting. (**e**) SKOV3 cells were treated with different concentrations of LicA for 24 h. Expression levels of p-mTOR, mTOR, p-4EBP1, 4EBP1, and eIF4E were analyzed by Western blotting. Data are presented as mean ± SEM. * *p* < 0.05, ** *p* < 0.01, and *** *p* < 0.001 compared with untreated control cells (0 μM).

## Data Availability

The data presented in this study are available on request from the corresponding author.

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
