# Peer review of "Licochalcone A Exerts Anti-Cancer Activity by Inhibiting STAT3 in SKOV3 Human Ovarian Cancer Cells"

_biomedicines, 2023, doi:10.3390/biomedicines11051264_

Round 1
Reviewer 1 Report
The manuscript entitled " Licochalcone A Exerts Anticancer Activity by Inhibiting STAT3 in Human Ovarian Cancer Cells " by JeongHyeon et al. studied the effects of Licochalcone A on proliferation, apoptosis, and cell cycle arrest of SKOV3 human epithelial ovarian cancer cells and identified possible mechanism involved in the anticancer activity of Licochalcone A against ovarian cancer through STAT3-mediation. It is interesting. However, the author needs to address the following comments before publishing this article in this journal. My comments are
- The introduction needs to be clear. The author should improve the according to the present study hypothesis and add novelty to the study.
- Licochalcone A is soluble in an organic solvent. The author should add how the drug was prepared and treated for experiments.
- The author displayed many results but could have discussed them more clearly. So, the author needs to rewrite the results and discussion section.
- The author should add the toxicity of Licochalcone A on normal cells, or it needs to be discussed.
- Did the author conduct any basic ovarian tumour experiments on the animal?
Author Response
The manuscript entitled " Licochalcone A Exerts Anticancer Activity by Inhibiting STAT3 in Human Ovarian Cancer Cells " by JeongHyeon et al. studied the effects of Licochalcone A on proliferation, apoptosis, and cell cycle arrest of SKOV3 human epithelial ovarian cancer cells and identified possible mechanism involved in the anticancer activity of Licochalcone A against ovarian cancer through STAT3-mediation. It is interesting. However, the author needs to address the following comments before publishing this article in this journal. My comments are
- The introduction needs to be clear. The author should improve the according to the present study hypothesis and add novelty to the study.
Thanks for your comments. According to the opinions of the reviewer, the introduction was modified to reveal the hypotheses and originality of the present study.
- Licochalcone A is soluble in an organic solvent. The author should add how the drug was prepared and treated for experiments.
We added methods for preparing and handling LicA for experiments in the revised manuscript (Method Section).
- The author displayed many results but could have discussed them more clearly. So, the author needs to rewrite the results and discussion section.
Thanks for your comments. Results and discussions were revised and supplemented in the revised manuscript according to the reviewer's comments.
- The author should add the toxicity of Licochalcone A on normal cells, or it needs to be discussed.
Thanks for the constructive advice. Indeed, our research team studied LicA angiogenesis using HUVEC cells in previous experiments and found that LicA was not cytotoxic to HUVEC cells up to 10 μM [A]. Based on this study, a discussion of toxicity to normal cells has been added in revised manuscript.
- Kim YH, Shin EK, Kim DH, Lee HH, Park JH, Kim JK. Antiangiogenic effect of licochalcone A. Biochem Pharmacol. 2010 Oct 15;80(8):1152-9. doi: 10.1016/j.bcp.2010.07.006.
- Did the author conduct any basic ovarian tumour experiments on the animal?
We are well aware of the absolute necessity of in vivo experiments using laboratory animals. We plan to do this in a follow-up study.
Reviewer 2 Report
Opinion
on the manuscript by JeongHyeon Seo et al., entitled "Licochalcone A Exerts Anticancer Activity by Inhibiting STAT3 in Human Ovarian Cancer Cells"
Manuscript ID: biomedicines-2271350
The paper by JeongHyeon Seo et al. deals with the anticancer properties of the natural product licochalcone A (LicA) on human ovarian cancer cells. The topic's relevance is high, and the Authors utilized generally accepted in vitro cell-based methods. The paper is well-written and contains only a few minor points to be corrected before publication. These are the following points:
Minor points:
1. The authors utilized six concentrations of the tested compound (fig. 1b), which seems efficient for calculating IC50 value. Therefore, fitting a sigmoid curve to resent the IC50 value of LicA is highly recommended.
2. Lines 140-141: "Expression levels of DR4 and FasL on the cell surface of SKOV3 cells were also reduced by LicA treatment as determined by flow cytometry (Figure 2b).". The cited results are presented in fig. 2c, and not in 2b.
3. Lines 117-118: "Comparisons were performed by one-way analysis of variance (ANOVA) or paired and unpaired t test when appropriate.". Since all statistical evaluations contain more than two groups, ANOVA is the only appropriate statistical method.
4. Column chart of Fig. 6d: all columns are statistically compared with untreated cells. However, cells treated with LicA and MG262 or CHX should be compared with LicA-treated cells also.
If these minor points are corrected, the acceptance of the manuscript is suggested.
Author Response
The paper by JeongHyeon Seo et al. deals with the anticancer properties of the natural product licochalcone A (LicA) on human ovarian cancer cells. The topic's relevance is high, and the Authors utilized generally accepted in vitro cell-based methods. The paper is well-written and contains only a few minor points to be corrected before publication. These are the following points:
- The authors utilized six concentrations of the tested compound (fig. 1b), which seems efficient for calculating IC50 value. Therefore, fitting a sigmoid curve to resent the IC50 value of LicA is highly recommended.
Following the constructive advice of the reviewer, the IC50 in Fig. 1b was obtained and presented as Supplementary Fig. 1.
- Lines 140-141: "Expression levels of DR4 and FasL on the cell surface of SKOV3 cells were also reduced by LicA treatment as determined by flow cytometry (Figure 2b).". The cited results are presented in fig. 2c, and not in 2b.
Sorry for this mistake. It's corrected in revised manuscript.
- Lines 117-118: "Comparisons were performed by one-way analysis of variance (ANOVA) or paired and unpaired t test when appropriate.". Since all statistical evaluations contain more than two groups, ANOVA is the only appropriate statistical method.
Sorry for this mistake. It's corrected in revised manuscript.
- Column chart of Fig. 6d: all columns are statistically compared with untreated cells. However, cells treated with LicA and MG262 or CHX should be compared with LicA-treated cells also.
Cells treated with LicA and MG262 or CHX were compared to LicA-treated cells, but no statistically significant differences were observed. However, this part has been added to the revised manuscript for clarity.
Reviewer 3 Report
The study is too preliminary. The authors have documented the anticancer action of licochalcone A using only one cancer cell line, namely SKOV3 human ovarian cancer cells. This is inadequate. More ovarian cancer cells must be used to exclude some private mechanisms observed only in SKOV3 cells. Can licochalcone A be considered as an inhibitor of STAT3 in also other ovarian cancer cells? This must be addressed. What about the action of licochalcone A against normal cells (up to 100 µM)? This must be evaluated. Please provide IC50 values for STAT3 and normal cells and compare them. This is essential to document selective anticancer action of licochalcone A.
The novelty of the study is limited. This is also stated by the authors. The anticancer action of licochalcone A was repeatedly reported in a number of cancer cell lines. Furthermore, it was already published that licochalcone A is a specific inhibitor for Stat3 (Funakoshi-Tago M, Tago K, Nishizawa C, Takahashi K, Mashino T, Iwata S, Inoue H, Sonoda Y, Kasahara T. Licochalcone A is a potent inhibitor of TEL-Jak2-mediated transformation through the specific inhibition of Stat3 activation. Biochem Pharmacol. 2008 Dec 15;76(12):1681-93. doi: 10.1016/j.bcp.2008.09.012). The authors did not cite this paper!
The role of STAT3 should be validated using up- and down-regulation approach.
Materials and methods: more information is needed. For example, please provide the concentrations of antibodies (western blotting).
Taken together, the study design is inadequate and there are numerous issues that must be addressed. Additional data are needed. Thus, rejection is recommended.
Author Response
The study is too preliminary. The authors have documented the anticancer action of licochalcone A using only one cancer cell line, namely SKOV3 human ovarian cancer cells. This is inadequate. More ovarian cancer cells must be used to exclude some private mechanisms observed only in SKOV3 cells. Can licochalcone A be considered as an inhibitor of STAT3 in also other ovarian cancer cells? This must be addressed. What about the action of licochalcone A against normal cells (up to 100 µM)? This must be evaluated. Please provide IC50 values for STAT3 and normal cells and compare them. This is essential to document selective anticancer action of licochalcone A.
We fully agree that it is appropriate to look at the antitumor activity of licochalcone A in multiple ovarian cancer cell lines rather than a single ovarian cancer cell line as advised by the reviewer. It was ideal to conduct research using various ovarian cancer cell lines, but in this study, SKOV3, an ovarian cancer cell line owned by the research team and one of the most used cell lines for ovarian cancer-related research by many researchers, was used. In future studies, we will verify the antitumor activity using various ovarian cancer cell lines and in vivo experiment.
To be honest, whether licochalcone A is an inhibitor of STAT3 in ovarian cancer cell lines other than SKOV3 and IC50 values of STAT3 activity in normal and cancer cells cannot be answered at present. We would appreciate it if the reviewers could understand that we cannot give an accurate answer to the above questions at this point, as the given revision period is too short to carry out these additional experiments, and it takes time to purchase cells and reagents. However, this will be clarified through additional experiments, and the data derived from it will be presented in another manuscript.
The novelty of the study is limited. This is also stated by the authors. The anticancer action of licochalcone A was repeatedly reported in a number of cancer cell lines. Furthermore, it was already published that licochalcone A is a specific inhibitor for Stat3 (Funakoshi-Tago M, Tago K, Nishizawa C, Takahashi K, Mashino T, Iwata S, Inoue H, Sonoda Y, Kasahara T. Licochalcone A is a potent inhibitor of TEL-Jak2-mediated transformation through the specific inhibition of Stat3 activation. Biochem Pharmacol. 2008 Dec 15;76(12):1681-93. doi: 10.1016/j.bcp.2008.09.012). The authors did not cite this paper! The role of STAT3 should be validated using up- and down-regulation approach.
As noted by the reviewers, Funakoshi-Tago et al. found that licochalcone A inhibited STAT3 phosphorylation and nuclear translocation in cells transduced with TEL-Jak2 into Ba/F3 cells. Unlike their study, however, in this study, licochalcone A inhibited STAT3 protein expression in SKOV3 cells, and the suppression of STAT3 expression was found to be due to the inhibition of translation initiation. We consider these observations to be the major uniqueness of the present study. We revised and added this part according to the meaningful comments of the reviewers in revised manuscript.
Materials and methods: more information is needed. For example, please provide the concentrations of antibodies (western blotting).
Following the constructive advice of the reviewer, provide the concentrations of antibodies in the revised manuscript.
Round 2
Reviewer 1 Report
The authors have satisfactorily responded to all comments and made the necessary changes to the manuscript.
Author Response
March 31, 2023
Editor-in-Chief
Biomedicines
Thank you for your letter dated March 30, 2023 (biomedicines-2271350R1), concerning the status of our manuscript entitled “Licochalcone A Exerts Anticancer Activity by Inhibiting STAT3 in Human Ovarian Cancer Cells." The authors deeply appreciate the reviewer’ constructive comments. We have revised the manuscript to incorporate these comments, as indicated in the following point-by-point responses.
Reviewer 1:
The authors have satisfactorily responded to all comments and made the necessary changes to the manuscript.
Thanks for your comments on the resubmission.
Reviewer 2:
The authors provided only insignificant corrections to their revised manuscript. The authors ignored my comments on adding data on more ovarian cancer cell lines and normal cells. It is very important to document specific and selective anticancer action of tested compound and exclude side effects on normal cells. The study is still preliminary. The authors also ignored my comments that the role of STAT3 should be validated using up- and down-regulation approach. Additional data are still required. If the authors are unable to provide new data, the manuscript should be rejected.
First of all, thank you for your sincere response to the resubmitted manuscript. We think there is a misunderstanding that needs to be clarified. We 100% agree that the effect of LicA should be validated in ovarian cancer cell lines other than SKOV3 and the role of STAT3 should be validated using up- and down-regulation approaches. This valuable opinion of the reviewer was never ignored. However, we do not have other ovarian cancer cell lines and normal ovarian epithelial cells, nor do we have reagents to perform up- and down-regulation approaches. It will take at least 2-3 weeks to purchase these cells and reagents and come to the lab, and it will take a lot of time to proceed and organize the experiments afterwards. It was 7 days given for re-submission, and it was impossible to edit all of this in 7 days. For this reason, we discussed the toxicity of LicA to normal cells by citing the toxicity results of HUVEC cells used in our previous studies in 1st revised manuscript. In addition, it was mentioned in the cover letter that the anticancer activity of LicA against other ovarian cancer cell lines and the role of STAT3 using up- and down-regulation approaches will be identified and reported through future studies. Once again, we would appreciate it if you could understand that We were not ignoring the reviewer's opinion, and there was such a complicated situation.
We would appreciate it if you could understand that we can only reply like this because the re-resubmission period is still given as 7 days. However, as mentioned above, this study will continue, and reviewers' comments are absolutely necessary to reveal the anticancer activity of LicA in ovarian cancer cell lines, so we will definitely conduct it and disclose it in a follow-up paper.
We would like to thank the editor and reviewer for their helpful remarks. We hope that our paper is now ready for publication in Biomedicines.
Sincerely yours,
However, as mentioned earlier, this research will continue and everything will be revealed and reported according to the opinions of the judges.
Jin-Kyung Kim, Ph.D.
Professor
Department of Biomedical Science,
Daegu Catholic University,
Gyeongsan-Si Gyeongbuk, Rep.of Korea 38430
E-mail: toto0818@cu.ac.kr

Reviewer 3 Report
The authors provided only insignificant corrections to their revised manuscript. The authors ignored my comments on adding data on more ovarian cancer cell lines and normal cells. It is very important to document specific and selective anticancer action of tested compound and exclude side effects on normal cells. The study is still preliminary. The authors also ignored my comments that the role of STAT3 should be validated using up- and down-regulation approach. Additional data are still required. If the authors are unable to provide new data, the manuscript should be rejected.
Author Response

(The authors gave the same response as above.)
